# Evaluation of Indigo Naturalis Prepared Using a Novel Method: Therapeutic Effects on Experimental Ulcerative Colitis in Mice

**DOI:** 10.3390/pharmaceutics17050674

**Published:** 2025-05-20

**Authors:** Xianxiang Xu, Lin Lin, Wenjie Ning, Xinyi Zhou, Aftab Ullah, Huiyong Yang, Xunxun Wu, Yong Diao

**Affiliations:** School of Medicine, Huaqiao University, 269 Chenghua North Road, Quanzhou 362021, China

**Keywords:** Indigo naturalis, Indigo Naturalis prepared using a novel method (NIN), ulcerative colitis, inflammation, aryl hydrocarbon receptor

## Abstract

**Background/Objectives**: Indigo naturalis (IN) is a traditional Chinese medicine concocted from medicinal plants such as *Baphicacanthus cusia* (Nees) Bremek. IN has multifaceted pharmacological activities. Recent research highlights the remarkable efficacy of IN in treating ulcerative colitis (UC). This study investigates the efficacy of Indigo Naturalis prepared using a novel method (NIN) in ameliorating UC. **Methods**: We have developed a new IN processing technology without the use of lime. Correspondingly, the content of active ingredients has relatively increased in NIN. In this study, dextran sulfate sodium salt (DSS) induced UC models among male KM mice, and the protective effects of NIN on UC were verified. **Results**: NIN could significantly improve weight loss, diarrhea and prolapse, bloody stools, elevated Disease Activity Index (DAI) and alleviate the colitis symptoms of mice; it could also improve the shortening of colon, disappearance of intestinal crypts, epithelial cell destruction and inflammatory infiltration caused by UC; and it could also significantly reduce the Histological Index (HI). In addition, NIN relieved the inflammatory response by decreasing the content of pro-inflammatory cytokines TNF-α and IL-1β and elevating the content of anti-inflammatory cytokines IL-10 and IL-22. It also restored the intestinal mucosal barrier by increasing the level of MUC2 protein expression at the site of colonic injury. **Conclusions**: The significant effects of NIN on UC were verified for the first time, suggesting that NIN was worth further developing into a novel therapeutic drug and, necessarily, further safety evaluations and comparisons with traditional IN will help in the application of NIN.

## 1. Introduction

Ulcerative colitis (UC) is a chronic inflammatory bowel disease (IBD), and its prevalence is increasing worldwide. UC is characterized by mucosal inflammation that extends continuously from the distal rectum to the proximal part and can occur in different locations of the colon, and long-term chronic inflammation can lead to abdominal pain, mucopurulent blood, and other symptoms [1,2]. The exact pathogenesis of UC has not yet been fully elucidated, Accumulating data have suggested that its pathogenesis is multifactorial, involving genetic predisposition, environmental factors, microbial dysbiosis, and dysregulated immune responses, multifactorial interactions of genetics, the environment, immunity, and intestinal microbial dysregulation [3,4,5,6,7]. The treatment of UC is still challenging.

Traditional Chinese Medicine (TCM) has shown great progress in the research of treating UC, with low side effects [8,9]. Indigo naturalis (IN), also called “Qingdai”, is processed from the stems and leaves of *Baphicacanthus cusia* (Nees) Bremek., *Polygonum tinctorium* Ait. and *Isatis indigotica* L. IN has been shown to ameliorate UC in clinical trials, even suggested to be more effective than 5-aminosalicylate [10,11,12]. Furthermore, indole compounds such as indigo and indirubin, the main ingredients in IN, act as ligands for the aryl hydrocarbon receptor (AhR). Activation of the AhR signaling pathway stimulates mucosal type 3 innate lymphoid cells to produce interleukin-22, which induces antimicrobial peptide and tight-junction molecule production [13,14,15].

Jian Qing-Dai (extracted from *Baphicacanthus cusia*. and produced in Fujian) is considered the highest-quality version of IN in China. Conventionally, the IN manufacturing method includes collecting stems and leaves, soaking lime and fermentation, producing coarse indigo, refining and drying [16]. The step of beating indigo in the concoction process tends to introduce a large amount of lime; Ca(OH)_2_ in lime reacts with CO_3_^2−^ in the soaking solution to form the CaCO_3_ nucleus, which provides carriers for adsorbing indigo, indirubin and other compounds. Simultaneously, commercial IN is mainly composed of 10% organic matter and 90% inorganic substance [17]. Inorganic impurities from lime are irritating to the digestive tract after consumption, and it contains less active ingredients.

Our team has developed a new IN processing method [18]. In short, lime is replaced by sodium hydroxide. So, in this way, the prepared IN does not contain any lime-like components. Simultaneously, the effective ingredients have greatly increased in the Indigo Naturalis prepared using a novel method (NIN). In this study, commonly used DSS-induced UC models were produced in mice; the effects of NIN on UC were investigated by evaluating the DAI score, colon length, and HI; the levels of inflammatory factors in UC mice were detected by ELISA; and the expression of the mucosal barrier-associated mucin MUC2 and the target protein AhR at the site of colonic injury was detected using immunohistochemistry.

## 2. Materials and Methods

### 2.1. Chemicals and Reagents

The stems and leaves of *Baphicacanthus cusia* were grown by the group, and standard IN was purchased from major pharmacies in Quanzhou City. N, N-dimethylformamide was purchased from Xilong (Shantou, China), and indirubin and indigo were purchased from Chengdu Mansion (Chengdu, China). Further, 4% paraformaldehyde general-purpose tissue fixative was purchased from White Shark (Hefei, China). TNF-α ELISA kits, IL-1β ELISA kits, IL-10 ELISA kits, IL-22 ELISA kist, and MUC2 Rabbit Polyclonal antibody were purchased from Proteintech (Wuhan, China). Occult blood detection kits were purchased from Nanjing Jiancheng (Nanjing, China). Hematoxylin, eosin, neutral gum, and PBS buffer dry powder were purchased from Solepol (Beijing, China). QuickBlockTM blocking solution immunostaining blocking solution, horseradish peroxidase-labeled goat anti-rabbit IgG (H+L), RIPA lysate (strong), and protease phosphatase inhibitor mixture were purchased from BiyunTian (Nantong, China). AhR Rabbit Polyclonal antibody was purchased from ABclonal (Wuhan, China). Powdered antigen repair solution (citrate method) and enhanced DAB Plus Kits were purchased from Fuzhou Meixin (Fuzhou, China). Carboxymethyl cellulose sodium (CMC-Na2), xylene, paraffin (soft wax) were purchased from Sinopharm (Shanghai, China). Mesalazine enteric-coated tablets were purchased from Losan Pharma GmbH (Neuenburg am Rhein Baden, Germany). Dextrose sodium sulfate (DSS, Colitis grade) was purchased from MP Biomedicals (Santa Ana, CA, USA).

### 2.2. Preparation of NIN and Determination of Active Ingredient Content

NIN was prepared via the concoction process developed in the laboratory in the early stage. The leaves from *Baphicacanthus cusia* were washed and dried and placed for a few days to be cut to 1–2 cm after the leaves were blackened, weighed 300 g and added 3 L of deionized water, extracted by refluxing at a constant temperature of 79 °C for 33 min, filtered to remove the dregs, and the filtrate was taken and added with 3.4% hydrochloric acid solution of 1 L and then hydrolyzed by heating in a water bath at 80 °C for 2 h. The pH value of the solution was adjusted to 6–8 with NaOH rather than Ca(OH)_2_ and then left to stand overnight. Then, centrifuge the next day at 7000 rpm room temperature for 5 min, take the washed precipitate to the evaporation dish, collect the supernatant and centrifuge again, and combine with the precipitate suspension in the evaporation dish, with 60 °C baking overnight. The next day, after grinding to a fine powder, NIN is obtained.

With reference to the optimal concentration of each sample and solution preparation method in the preconstructed HPLC assay method [19], the concentrations of indigo and indirubin control solution and NIN, SIN test solutions were prepared, so that the concentrations were 25, 50, 100, 2000 μg/mL, respectively. The HPLC chromatographic conditions were as follows: Elite YWG C18 (4.6 × 200 mm) chromatographic column, with a 4.6 × 200 mm HPLC column. The HPLC conditions were as follows: Elite YWG C18 (4.6 × 200 mm) column, methanol/water (7:3) as mobile phase, flow rate of 1 mL/min, detection wavelength of 292 nm.

### 2.3. Modeling and Grouping

The care and experimental protocol involving KM mice was conducted in accordance with the ARRIVE guidelines and approved by the Ethics Committee for Experimental Animal Management of Huaqiao University (A2019036). Thus, 60 male KM mice weighing 20 ± 2 g were purchased from Beijing Huafukang Biological Science and Technology Co. (Beijing, China). All experimental mice were kept in independent mouse room with air conditioning (free feeding and free lighting). Referring to Stefan’s method [20] and our pre-experiment, DSS with molecular weight of 36–50 kD was used to formulate 4% DSS (*w*/*w*) solution with deionized water, replacing drinking water so that the mice were free to drink it for 12 days, and then changed to free to drink deionized water on the 13th and 14th days. DSS-induced mice are used as UC models.

Experimental grouping and dosing regimen: 60 KM mice were selected by random number table and randomly divided into a normal control group (CON), a model group (MOD), a low-dose group of the new process IN (LIN), a high-dose group of the new process IN (HIN), a positive control group (POS), and a control group of the commercially available standard IN (SIN) after 7 days of acclimatization, with 10 randomly numbered mice in each group. Daily in the 14-day experimental process, CON and MOD were given 0.5% CMC-Na_2_, LIN and HIN were given 200 and 400 mg/kg of NIN test solution, POS was given 200 mg/kg Mesalazine (5-aminosalicylic acid) test solution, and SIN was given 400 mg/kg test solution. The volume of gavage for each group was 0.2 mL/10 g. Only the investigator was aware of the group allocation at the different stages of the experiment.

### 2.4. Assessment of Disease Activity Index (DAI) in Mice

The rate of change in body weight of mice was recorded and scored daily. Fresh feces of mice were collected daily and scored after observing the traits, and after scoring, the collected feces were tested using an occult blood test kit with reference to the steps specified in the instructions. Disease Activity Index (DAI) scoring (Table 1) was performed with reference to the scoring criteria of Cooper [21], DAI = (weight loss score + fecal trait score + fecal occult blood score)/3.

### 2.5. Collection and Processing of Mouse Colon Specimens

At the end of the experiment, the mice were subjected to de-verbalized execution, and all the intestinal tubes from the cecum to the anus were dissected and taken. The cecum was used to mark the beginning of the specimen, and the length of the specimen was recorded and photographed. Subsequently, the cecum was cut off, the intestinal tract was rinsed with pre-cooled PBS, and the proximal cecum section was placed in an ultra-low-temperature refrigerator at −80 °C to be preserved for subsequent testing.

The proximal anal segment was fixed in 4% paraformaldehyde tissue fixative for 24 h at room temperature and then removed and rinsed with running water to remove the residual fixative on the surface. After being cut into 3 mm colon segments, the tissues were dehydrated sequentially in gradient alcohol using an automatic tissue dehydrator, then immersed in xylene for clearing, and then infiltrated in paraffin wax to replace the xylene in the tissues. The colon tissue was prepared into paraffin blocks using a tissue embedding machine and sliced into 4 μm thick sections using a tissue slicer.

### 2.6. Assessment of HI

The prepared 4 μm thick paraffin sections were baked at 60 °C for 2 h, followed by dewaxing, hydration, hematoxylin staining for 8 min and washed with running water to remove the floating color on the surface, then differentiated using 1% hydrochloric acid-alcohol differentiation solution for 2 s, and washed with running water to remove the residue. Subsequently, they were immersed in tap water at 45 °C for 15 min, stained with eosin for 2 min, and finally treated with gradient alcohol dehydration and xylene for transparency, neutral gum sealing, and microscopic observation and image acquisition. The H&E-stained images were evaluated for HI (Table 2), with reference to the histological scoring rules for the mouse intestinal inflammation model [22], HI = Score 1 + Score 2.

### 2.7. ELISA

Remove the colon from −80 °C ultra-low-temperature refrigerator; after melting on ice, weigh and it cut into pieces and place them in the pre-cooled centrifuge tube on ice, add the lysate at a ratio of 1 mL of lysate to every 0.1 g of tissue, homogenize thoroughly, centrifuge the homogenate at 4 °C for 10 min at 12,000× *g*, take the supernatant, then the samples to be tested, and then store them in separate packages at −80 °C. Referring to the instructions of the ELISA kits of Wuhan Three Eagles Company, the test was carried out after the pre-test of dilution of each index tissue homogenate sample.

### 2.8. Immunohistochemistry

Immunohistochemical staining of colon tissues: Briefly, 4 μm thick paraffin sections were sequentially baked, deparaffinized and hydrated, and then incubated with drops of 3% H_2_O_2_ at room temperature. The sections were then repaired in an antigen repair autoclave, and the tissues were sealed with immunostaining sealing solution at room temperature and then incubated overnight at 4 °C with a droplet of primary antibody. The details of the antibodies used are shown in Table 3. After PBST immersion, a droplet of HRP-labelled secondary antibody was added and incubated at room temperature. The tissue was washed again with PBST, and then DAB chromogenic solution was added to the tissue at room temperature for 4 min. Hematoxylin re-staining-alcoholic differentiation in hydrochloric acid returning to blue in tap water. Dehydration—Transparency—Seal the film, observe under the microscope and collect images.

Protein expression assessment: Immunohistochemical staining images were analyzed using Image-Pro Plus 6.0 software. The area (Area) was measured by circling the complete colonic structures at 100× field of view, and the Area Sum value was collected using the data collector; brown pixel points representing positive protein expression were selected using the color separation selection tool, the integrated optical density (IOD) of the brown pixel points within the circled area was measured, and the IOD Sum value was collected using the data collector, AOD = IOD Sum/Area Sum.

### 2.9. Statistical Analysis

GraphPad Prism 6 was used for graphing, and all data were expressed as mean ± standard deviation. SPSS 18.0 software was used to first test the normality of the raw data and then perform one-way ANOVA for the continuous variables that obeyed normal distribution, followed by post hoc test for chi-square hypothesis. We then used Welch’s corrected *t*-test for two independent samples to evaluate the significance of the parameters between the groups if the variance was not equal; for the scores that did not obey normal distribution, non-parametric tests were used to evaluate the significance of the parameters between the groups, and two-by-two comparisons were made between the scores of the data. For scores that did not follow a normal distribution, non-parametric tests were used to evaluate the significance of the parameters between groups, with the Mann–Whitney test for two-by-two comparisons and the Kruskal–Wallis test for multiple comparisons.

## 3. Results

### 3.1. Appearance and HPLC Fingerprint of NIN

NIN properties are consistent with the pre-laboratory product, a blue-black powder, light in mass, easy to fly, and with a faint odor. The color of NIN is much darker than traditional SIN (Figure 1).

From peak number, peak area, and peak shape, the fingerprint of NIN has great similarity with the fingerprint of SIN in HPLC (Figure 2). The retention time of indirubin and indigo in NIN was the same as that of SIN, and both of them were consistent with the standards. According to the standard curve, the percentages of indirubin and indigo in NIN were 29.24% and 15.63%, respectively, while the percentages of indirubin and indigo in SIN were 0.33% and 2.85%, respectively. Through UPLC-Q-TOF-MS/MS analysis, there are nine common components, including valine, (C_5_H_11_NO_2_, *m*/*z* 118.0862), isatin (C_8_H_5_NO_2_, *m*/*z* 148.0393), tryptanthrin (C_15_H_8_N_2_O_2_, *m*/*z* 249.0659), indigo (C_16_H_10_N_2_O_2_, *m*/*z* 263.0815), indirubin (C_16_H_10_N_2_O_2_, *m*/*z* 263.0815), hydroxyindirubin (C_16_H_10_N_2_O_3_, *m*/*z* 279.0764), 5,7,4’-trihydroxy-8-methoxyflavone (C_16_H_12_O_6_, *m*/*z* 301.0707), isorhamnetin (C_16_H_12_O_7_, *m*/*z* 317.0656), 2-(1H-indol-3-yl)acetamide (C_10_H_10_N_2_O, *m*/*z* 175.0866).

### 3.2. NIN Significantly Improves Colitis Symptoms Caused by UC in Mice

In order to investigate the protective effect of NIN on UC mice, the weight loss, fecal characteristics and fecal occult blood of UC mice were scored, and the DAI was calculated, as shown in Figure 3, which revealed that the weight loss, fecal dysmorphia and fecal occult blood of UC mice in each group were alleviated to varying degrees, of which the general conditions of the HIN and POS mice were significantly improved, their movements were more flexible than those of the MOD mice, their feces were shaped, no mice showed prolapse, their appetite was increased, their fecal occult blood and anal bleeding were significantly reduced, and their appetite was increased. HIN and POS mice showed significant improvements in general condition, more flexible movement than MOD mice, gradually shaped feces, and no mice showed prolapse of the anus. Occult blood in feces, bloody stools in the eyes, and anal bleeding were significantly weakened, and their appetites increased and weight loss slowed down. From the 10th day of the experiment, DAI showed a decreasing trend. These results indicate that NIN has a protective effect against ulcerative colitis in mice.

### 3.3. NIN Inhibits Colonic Shortening and Significantly Reduces HI in Mice

In colitis model experiments, colon length is the most intuitive way to judge the condition, and colon length shortens as the condition worsens. To further investigate the protective effect of NIN on UC mice, colon length and histopathological damage of UC mice were examined. The colon length of MOD mice was significantly shortened, and compared with MOD, HIN, POS, and SIN, all significantly inhibited colon shortening, and LIN significantly inhibited shortening of the colon in mice (Figure 4).

Compared with MOD, the scope of colonic mucosal lesions in each administration group was reduced, the degree of damage was reduced, the degree of reduction in cup cells was reduced and the crypts were repaired, among which the mucosal inflammatory cell infiltration of HIN and SIN was milder, the mucosal epithelium was basically intact, the crypt structure was restored, the lamina propria glands were restored to tubular, and the intestinal structure was basically restored to normal and intact (Figure 5). Compared with CON, colon HI was significantly elevated in MOD mice; compared with MOD, each administration group could improve colon HI, HIN and SIN improved better and could significantly reduce colon HI in UC mice; LIN and POS could reduce colon HI (Figure 6).

### 3.4. NIN Improves Inflammatory Response by Regulating the Levels of Inflammatory Factors in Mouse Colon Tissues

To investigate the effect of NIN on inflammatory factors in the colonic tissues of UC mice, the levels of pro-inflammatory cytokines TNF-α and IL-1β and inflammation-inhibiting cytokines IL-10 and IL-22 were detected in the colonic tissues using ELISA. The levels of TNF-α in the tissues of MOD mice were significantly increased, and the levels of TNF-α in the respective administered groups were significantly decreased (Figure 7a). IL-1β levels were significantly increased in the tissues of MOD mice compared to CON; IL-1β levels were significantly decreased in LIN, HIN, and SIN compared to MOD, while there was no significant change in POS (Figure 7b). Compared with CON, IL-10 levels were significantly decreased in tissues of MOD mice; IL-10 levels were significantly increased in all dosing groups compared with MOD; and IL-10 levels were significantly higher in HIN compared with SIN (Figure 7c). Compared with CON, IL-22 levels were significantly reduced in tissues of MOD mice; compared with MOD, LIN and SIN significantly increased IL-22 levels; HIN and POS significantly increased IL-22 levels; compared with SIN, IL-22 levels of HIN were significantly higher than those of SIN (Figure 7d).

### 3.5. NIN Restores the Intestinal Mucosal Barrier by Increasing MUC2 Protein Expression Level

In order to investigate the effect of NIN on the expression of mucin MUC2 associated with the mucosal barrier at the site of colonic injury in mice, the mouse colon was stained for MUC2 protein by IHC (Figure 8). The results show that MUC2 protein expression was significantly increased in MOD mice compared to CON, MUC2 protein expression was significantly increased in LIN and HIN mice, MUC2 protein expression was increased in SIN mice, and MUC2 protein expression was similar in POS and MOD, with no significant increase in the level of MUC2 protein expression. Protein expression was significantly increased in LIN and HIN mice compared with MOD, MUC2 protein expression was increased in SIN mice, and the levels of MUC2 protein expression in POS and MOD mice were similar and not significantly different; there were significant differences in the levels of MUC2 protein expression in LIN and HIN mice compared with SIN (Figure 9).

### 3.6. NIN Increases AhR Protein Expression at the Site of Colonic Injury in UC Mice

AhR, as a key regulatory molecule of cell signaling pathways, plays an important role in a variety of important biological processes in the organism. To further investigate the role of the new process IN on AhR protein expression at the site of colonic injury in UC mice, the colonic AhR protein of mice was stained by IHC (Figure 10), and the results showed a significant increase in the expression of MUC2 protein in MOD mice compared to CON, a significant increase in the expression of AhR protein in HIN mice and an increase in the AhR of SIN compared to MOD, and LIN, POS and MOD AhR protein expression levels were not significantly different (Figure 11).

## 4. Discussion

The appearance properties and qualitative identification of active ingredients of NIN were consistent with those of the original process, and the retention times of indirubin and indigo in NIN were consistent with those of the standard products. The calculated percentages of indirubin and indigo were significantly higher than those of traditional IN, and the content of indirubin was higher than that of the original process in the laboratory. It provided sufficient and high-quality raw material for the later experiments.

Based on the existing reports of the effects of IN on UC, the present experiment was conducted to verify the consistency of the anti-inflammatory effects of NIN and SIN by constructing a mouse UC model. In this study, DSS-induced UC mice showed increased DAI scores and shortened colons. The histopathological damage was aggravated, with a significant increase in the pro-inflammatory cytokines TNF-α and IL-1β and a significant decrease in the anti-inflammatory cytokines IL-10 and IL-22 in the colon tissue. All these demonstrated the successful replication of the DSS-induced mouse UC model. Relative to MOD, HIN and SIN significantly ameliorated weight loss, diarrhea and prolapse, bloody stools in the naked eye, and elevated DAI in mice caused by UC, suggesting that, in agreement with the effect of SIN, NIN alleviated UC disease symptoms. Secondly, LIN, HIN and SIN could improve the shortening of the colon, absence of crypts, destruction of epithelial cells and inflammatory infiltration in mice caused by UC and could significantly reduce colonic HI, suggesting that NIN could alleviate the pathological damage to the colon. LIN, HIN, and SIN could decrease the content of pro-inflammatory factors TNF-α and IL-1β and increase the content of anti-inflammatory factors IL-10 and IL-22 in the colonic tissues of UC mice, and the effect of NIN was consistent with that of SIN, which could alleviate the symptoms of UC. The effect of NIN on IL-1β and IL-10, IL-22 was significantly better than that of SIN. Compared to SIN, NIN showed superior efficacy in histological index improvement and cytokine modulation.

The intestinal mucosal barrier consists of the mucus layer of the intestinal wall and a mechanical barrier, and MUC2 secreted by cuprocytes is a major component of the mucus layer. In experimental colitis models, the induction of DSS destroys cuprocytes, reduces Muc2 mRNA expression, decreases the thickness of the mucus layer in the intestine, and increases the permeability of the intestine to bacteria [23,24]. In this study, MUC2 expression was significantly increased at the MOD injury, and HE staining showed that the MOD colon was extremely severely damaged, with extensive ulceration and granulation tissue formation. The formation of granulation tissue signifies that the organism has begun compensatory autonomous repair at the site of injury, so unlike previous literature reports, MUC2 protein expression was significantly increased at the site of MOD injury compared to the normal group. However, compared with MOD, NIN promoted MUC2 expression more significantly and better than SIN, suggesting that NIN can repair colonic mucosal injury in UC mice, promote mucus secretion from cup cells, and restore the self-protective function of intestinal mucosa.

AhR is a ligand-dependent receptor associated with the biological response to aromatic hydrocarbons. The activation of the AhR signaling pathway has an inhibitory effect on inflammation, which can be evoked by indole compounds in IN [25]. Therefore, in this study, we investigated whether NIN relieves UC symptoms and promotes mucosal repair in correlation with AhR by detecting AhR protein expression. In this study, AhR expression at the site of MOD injury was significantly increased compared with CON, which may be based on the body’s own mucosal protection mechanism and immune response regulation in experimental animals, which compensated for the increase in AhR expression, so that the damaged mucosa began to compensate for the autonomous repair. Compared with MOD, HIN and SIN increased AhR protein expression more significantly. It suggests that NIN alleviates UC symptoms, regulates inflammatory factor levels, and promotes colonic mucosal repair in mice by agonizing AhR.

In conclusion, our results indicate that NIN can significantly improve the symptoms of colitis caused by UC in mice, reduce colonic tissue damage, improve inflammatory response, and promote mucosal repair. A preliminary investigation of NIN against UC revealed that its anti-UC effect was related to the agonism of AhR. Further in-depth research is needed to explain the potential medical benefits and features of NIN. Future studies will explore the mechanism of action of NIN against UC from the perspective of inflammatory cancer transformation.

## Figures and Tables

**Figure 1 pharmaceutics-17-00674-f001:**
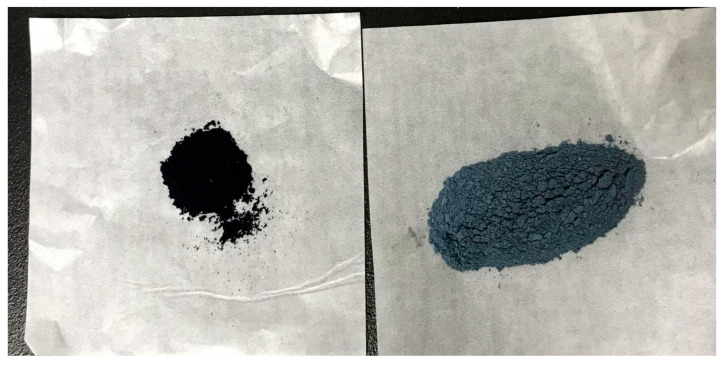
Appearance of NIN and SIN.

**Figure 2 pharmaceutics-17-00674-f002:**
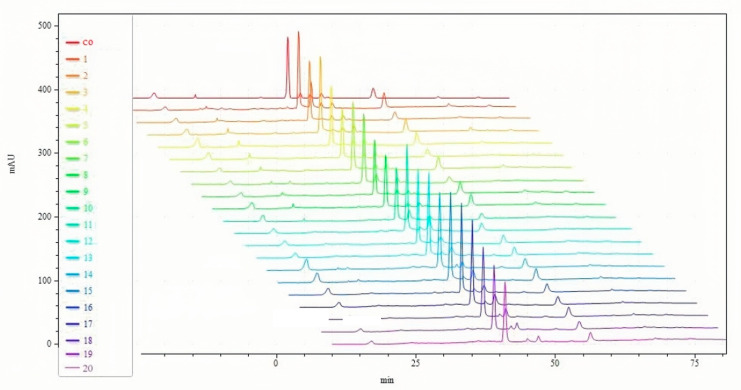
HPLC fingerprints of NIN and SIN. co: common; (1~10): SIN; (11~20): NIN.

**Figure 3 pharmaceutics-17-00674-f003:**
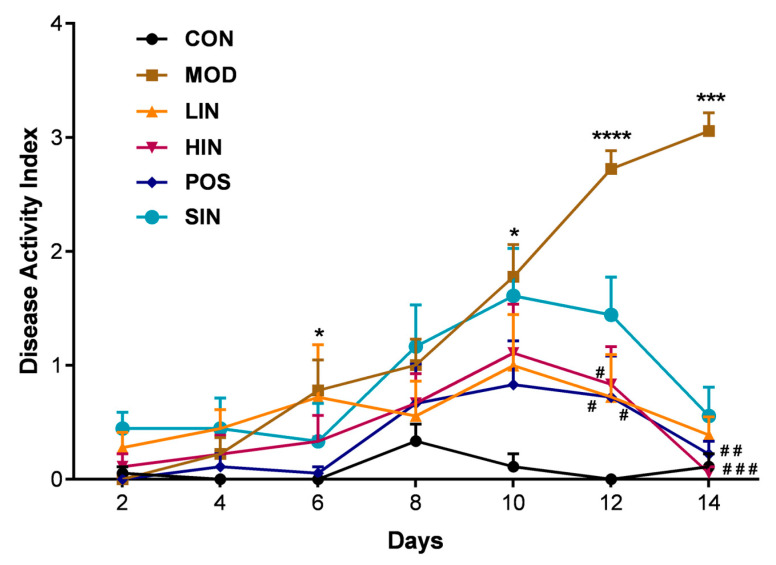
Effect of NIN on Disease Activity Index (DAI) in UC mice. *n* = 5, * *p* < 0.05, *** *p* < 0.001, **** *p* < 0.0001 vs. the normal control. # *p* < 0.05, ## *p* < 0.01, ### *p* < 0.001, vs. the model.

**Figure 4 pharmaceutics-17-00674-f004:**
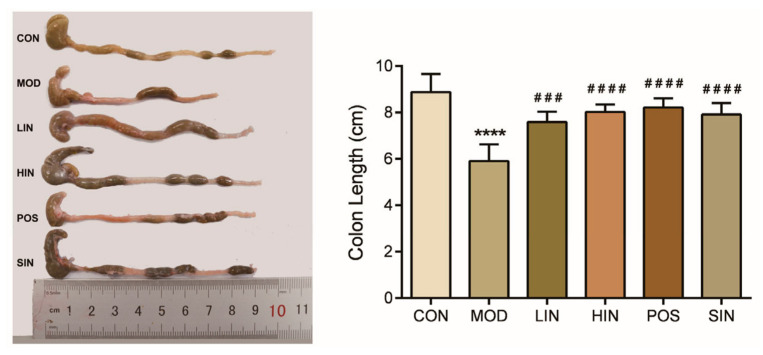
Effect of NIN on colon length in UC mice. *n* = 5, **** *p* < 0.0001 vs. the normal control. ### *p* < 0.001, #### *p* < 0.0001 vs. the model.

**Figure 5 pharmaceutics-17-00674-f005:**
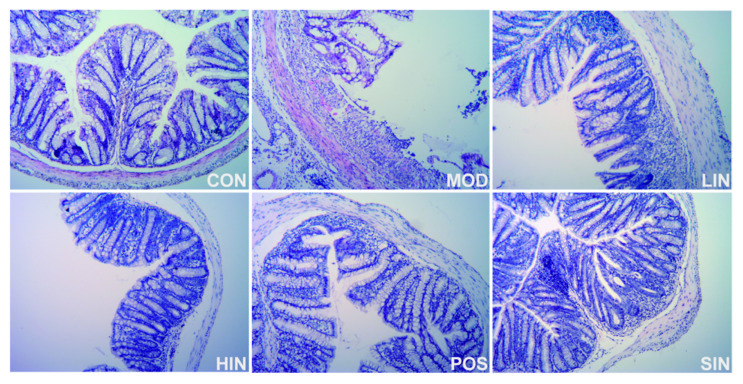
H&E staining of paraffin sections of colon of mice in each group. Representative images (×400) of colonic sections stained with H&E for histopathologic evaluations.

**Figure 6 pharmaceutics-17-00674-f006:**
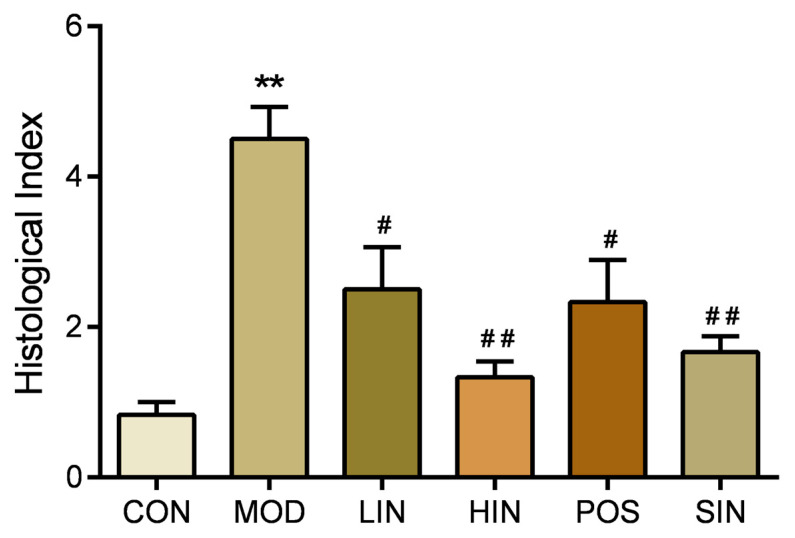
Effect of NIN on HI in UC mice. HI (tissue damage index) assessed by H&E staining of paraffin sections of mouse colon. *n* = 5, ** *p* < 0.01 vs. the normal control. # *p* < 0.05, ## *p* < 0.01 vs. the model.

**Figure 7 pharmaceutics-17-00674-f007:**
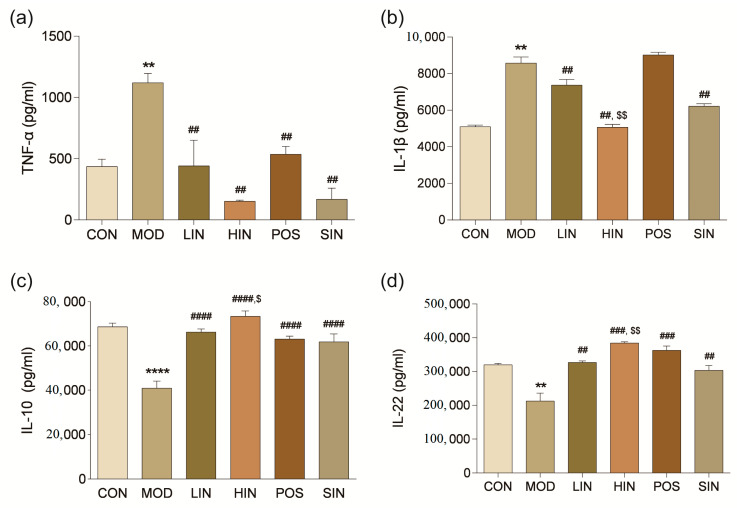
Effect of NIN on the levels of colonic inflammatory factors in UC mice. (**a**) TNF-α; (**b**) IL-1β; (**c**) IL-10; (**d**) IL-22. The effect of NIN on TNF-α, IL-1β, IL-10 and IL-22 was evaluated by ELISA. *n* = 5, ** *p* < 0.01, **** *p* < 0.0001 vs. the normal control. ## *p* < 0.01, ### *p* < 0.001, #### *p* < 0.0001 vs. the model. $ *p* < 0.05, $$ *p* < 0.01 vs. the traditional process IN.

**Figure 8 pharmaceutics-17-00674-f008:**
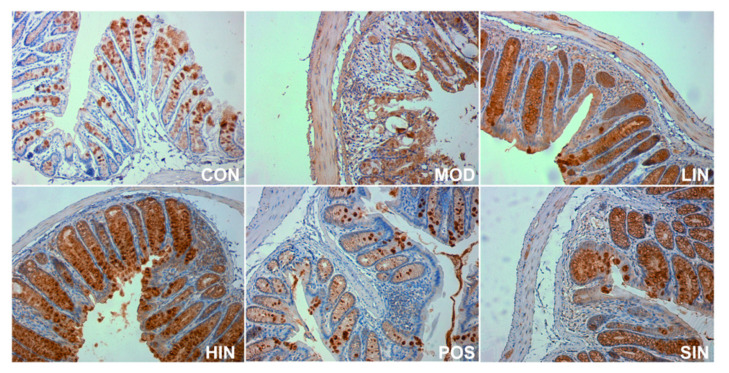
IHC staining of MUC2 protein in paraffin sections of mouse colon in each group (×400). The expression level of MUC2 protein was detected by immunohistochemistry.

**Figure 9 pharmaceutics-17-00674-f009:**
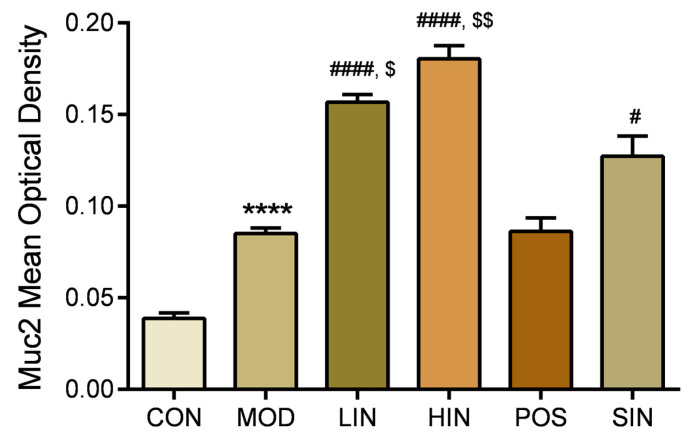
Mean optical density analysis of MUC2 protein IHC staining in paraffin sections of mouse colon in each group. MUC proteins were detected by performing average optical density (AOD) calculations on IHC-stained pictures. *n* = 5, **** *p* < 0.0001 vs. the normal control. # *p* < 0.05, #### *p* < 0.0001 vs. the model. $ *p* < 0.05, $$ *p* < 0.01 vs. the traditional process IN.

**Figure 10 pharmaceutics-17-00674-f010:**
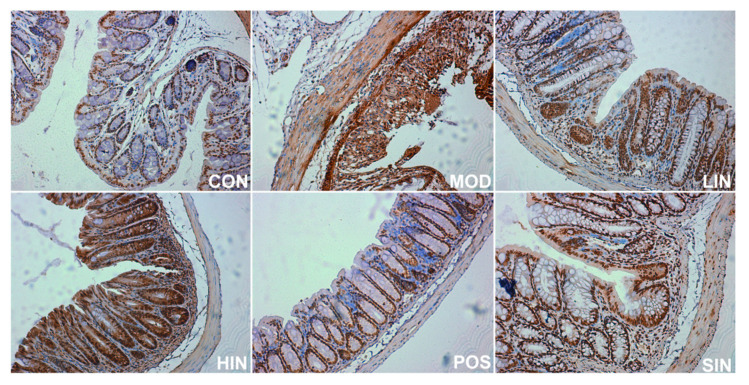
IHC staining of AhR protein in paraffin sections of mouse colon, by group (×400). The expression level of AhR protein was detected by immunohistochemistry.

**Figure 11 pharmaceutics-17-00674-f011:**
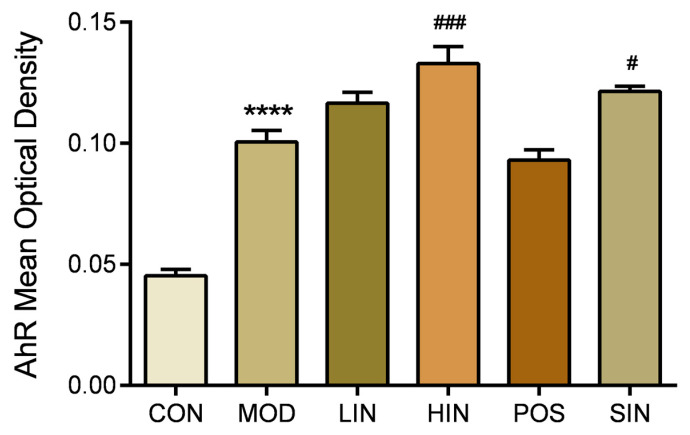
Mean optical density analysis of AhR protein IHC staining in paraffin sections of mouse colon in each group. AhR proteins were detected by performing average optical density (AOD) calculations on IHC-stained pictures. *n* = 5, **** *p* < 0.0001 vs. the normal control. # *p* < 0.05, ### *p* < 0.001 vs. the model.

**Table 1 pharmaceutics-17-00674-t001:** DAI scoring rules.

Symptoms	0 Points	1 Points	2 Points	3 Points	4 Points
Decreased body mass (%)	No	1–5	5–10	10–15	>15
Stool properties	Normal	—	Semi-dilute stools	—	loose stool
Blood in stool	Negative, weakly positive	—	Positive, strongly positive	—	bloody stools

**Table 2 pharmaceutics-17-00674-t002:** HI scoring rules.

Inflammatory Cell Infiltration			Intestinal Structure		
Severity Level	Degree of Infiltration	Rating 1	Epithelial Changes	Mucosal Structure	Rating 2
Mild (10–25%)	Mucosal layer	1	Localised erosion		1
Moderate (26–50%)	Mucosal layer and submucosal layer	2	celiac disease	±localised ulceration	2
Severe (>51%)	Transmural	3	celiac disease	Extensive ulceration ± granulation tissue ± pseudopolyp	3
Rating 1 + Rating 2					0–6

**Table 3 pharmaceutics-17-00674-t003:** List of immunohistochemical staining antibodies.

Antibody Name	Dilution Times	Antibody Source (Stock Number)
AhR (D5S6H) Rabbit mAb	1:1000	CST (#83200)
Non-phospho (Active) β-catenin (Ser33/37/Thr41) (D13A1) Rabbit mAb	1:1000	CST (#8814)
β-actin Rabbit Monoclonal Antibody	1:1000	Beyotime (AF5003)
Horseradish Peroxidase Labelled Goat Anti-rabbit IgG (H+L)	1:50	Beyotime (A0208)

## Data Availability

The data are available from the corresponding author on reasonable request.

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
