# Peer review of "Evaluation of Indigo Naturalis Prepared Using a Novel Method: Therapeutic Effects on Experimental Ulcerative Colitis in Mice"

_pharmaceutics, 2025, doi:10.3390/pharmaceutics17050674_

Round 1
Reviewer 1 Report
Comments and Suggestions for Authors
Xu and coauthors report a new method of preparation of Indigo Naturalis extraction from Baphicacanthus cusia leaves and its in vivo pharmacological and mechanistic studies of the extracted material in dextran sulfate sodium salt induced ulcerative colitis mice models. The authors concluded the newly developed extracting method led to enrichment of two proposed active components indirubin and indigo and improved disease symptoms as well as inflammatory biomarker profiles.
Several comments are listed for consideration:
- In section 2.2, "NIN was prepared by the concoction process developed by the laboratory in the early stage,... and extracted by refluxing at a constant temperature of 79 °C for 33 min, filtered to remove the dregs and the filtrate was taken and added with 3.4 % hydrochloric acid solution of 1 L, and then hydrolysed by heating in a water bath at 80 °C for 2 h. The pH value of the solution was adjusted to 6-8 and then left to stand overnight..." The newly developed method of extraction appears to use acid for hydrolysis, is there a rationale on this? What's the relationship between this acid treatment and sodium hydroxide mentioned in the last paragraph of the introduction?
- Figure 2. The authors have concluded the active components are enriched in the newly developed methods based on HPLC, is there a confirmation with mass spec or NMR, to confirm the molecules' identity. Also are the extraction method also enriches something else based on total HPLC profile that's potential safety concern?
- Figures 9-11, the authors suggested the more significant increase in Muc2 and AhR2 are beneficial in high and low dose regime of current newly developed extracts "NIN" vs. control group in the dextran sulfate sodium salt induced ulcerative colitis mice models. If this is beneficial during self-repair and healing of the tissues based on other physiological indices presented in the current study as shown in Figure 3, will there be follow-up recovery data to show Muc2 and AhR2 will change back to normal level as the control group beyond 14 days to suggest that the high and lose dose "NIN" treated disease model mice will indeed fully recover?
- Section 2.3, is the dosing of mice daily amount throughout 1-14 days?
- Minor comment, Is current study showing major advancement compared to previous studies in mice models in potential medical benefit over existing treatment of relevant disease indications?
Author Response
Thank you for your careful criticisms. We have made corresponding corrections highlighted in the manuscript and believe that all the changes could fulfil the requirements this time.
Revision notes, point-to-point, are given as follows:
#reviewer1:
- In section 2.2, "NIN was prepared by the concoction process developed by the laboratory in the early stage,... and extracted by refluxing at a constant temperature of 79 °C for 33 min, filtered to remove the dregs and the filtrate was taken and added with 3.4 % hydrochloric acid solution of 1 L, and then hydrolysed by heating in a water bath at 80 °C for 2 h. The pH value of the solution was adjusted to 6-8 and then left to stand overnight..." The newly developed method of extraction appears to use acid for hydrolysis, is there a rationale on this? What's the relationship between this acid treatment and sodium hydroxide mentioned in the last paragraph of the introduction?
Re: The purpose of hydrolysis is to hydrolyze the indigo contained in plants into indole, which is then oxidized and condensed to form indigo. The pH value of the solution was adjusted to 6-8 with NaOH rather than conventional Ca(OH)2. NaOH is added in the manuscript.
- Figure 2. The authors have concluded the active components are enriched in the newly developed methods based on HPLC, is there a confirmation with mass spec or NMR, to confirm the molecules' identity. Also are the extraction method also enriches something else based on total HPLC profile that's potential safety concern?
Re: The fingerprint of NIN has great similarity with the fingerprint of SIN in HPLC. They have nine common components by UPLC-Q-TOF-MS/MS analysis. The molecules' identity is added in the manuscript.
- Figures 9-11, the authors suggested the more significant increase in Muc2 and AhR2 are beneficial in high and low dose regime of current newly developed extracts "NIN" vs. control group in the dextran sulfate sodium salt induced ulcerative colitis mice models. If this is beneficial during self-repair and healing of the tissues based on other physiological indices presented in the current study as shown in Figure 3, will there be follow-up recovery data to show Muc2 and AhR2 will change back to normal level as the control group beyond 14 days to suggest that the high and lose dose "NIN" treated disease model mice will indeed fully recover?
Re: Unlike previous literature reports, the expression of MUC2 and AhR at the site of MOD injury is significantly increased in our study, which may be based on the mucosal protective mechanism and immune response regulation of the experimental animal body. However, NIN have a more significant and superior effect on promoting the expression of MUC2 and AhR, indicating that NIN can repair colonic mucosal damage in UC mice, promote goblet cell secretion of mucus, and restore intestinal mucosal self-protection function. As shown in Figure 3, HIN mice showed significant improvement in general condition, gradually shaped faeces, and no mice showed prolapse of the anus.
- Section 2.3, is the dosing of mice daily amount throughout 1-14 days?
Re: All groups of mice were administered daily or with control solution the 14 day experimental process. The description of administration method has been revised in the manuscript.
- Minor comment, Is current study showing major advancement compared to previous studies in mice models in potential medical benefit over existing treatment of relevant disease indications?
Re: Compared to SIN, NIN showed superior efficacy in histological index improvement and cytokine modulation. Still, potential medical benefit of NIN need further confirmation.
Reviewer 2 Report
Comments and Suggestions for Authors
Evaluation of Indigo Naturalis Prepared Using a Novel Method: Therapeutic Effects on Experimental Ulcerative Colitis in Mice
The authors describe new and relevant information about therapeutic effects on ulcerative colitis in mice, using a new method of preparation of Indigo Naturalis (IN) from leaves of Baphicacanthus cusia.
The experimental assays have been well designed, described and performed. The conclusions are supported by the experimental results.
The Manuscript is acceptable with minor modifications, as detailed below.
MINOR MODIFICATIONS
Figure 2
The chromatograms of Figure 2 are useful for readers of the Journal, since they illustrate the different composition of the extract obtained by the authors (NIN), regarding to that standard and commercial extract (SIN).
However, the chromatograms of Figure 2 have poor graphical quality. They are like screen captures of the chromatograms, with low resolution limited by the monitor pixels available.
Recommended Modification
- Export chromatograms coordinates in TXT, CSV, XLS or other format of tabulated data.
- Import the above files of tabulated data with plotting software of high resolution, such as SigmaPlot, GraphPad Prism or alternative programs.
- Export the new quality graphs with high resolution, 300 or 600 dots per inch, in TIF, PNG or another image format, as required by the journal.
Alternative Modification
If the Recommended modification is not possible for authors, I suggest to remove the Figure 2 from the main Manuscript, and to incorporate the current Figure 2 as Supplementary Material file (PDF, DOC, TIF, PNG, or another format).
Then, the authors must renumber the Figures and check the text of the main Manuscript, as well as the new Figure S1 of the Supplementary Material file.
Best regards.
Author Response
Dear Dr:
Thank you for your careful criticisms. We have made corresponding corrections highlighted in the manuscript and believe that all the changes could fulfil the requirements this time.
Revision notes are given as follows:
#reviewer2:
The authors describe new and relevant information about therapeutic effects on ulcerative colitis in mice, using a new method of preparation of Indigo Naturalis (IN) from leaves of Baphicacanthus cusia. The experimental assays have been well designed, described and performed. The conclusions are supported by the experimental results. The Manuscript is acceptable with minor modifications, as detailed below. MINOR MODIFICATIONS Figure 2 The chromatograms of Figure 2 are useful for readers of the Journal, since they illustrate the different composition of the extract obtained by the authors (NIN), regarding to that standard and commercial extract (SIN). However, the chromatograms of Figure 2 have poor graphical quality. They are like screen captures of the chromatograms, with low resolution limited by the monitor pixels available. Recommended Modification Export chromatograms coordinates in TXT, CSV, XLS or other format of tabulated data. Import the above files of tabulated data with plotting software of high resolution, such as SigmaPlot, GraphPad Prism or alternative programs. Export the new quality graphs with high resolution, 300 or 600 dots per inch, in TIF, PNG or another image format, as required by the journal. Alternative Modification If the Recommended modification is not possible for authors, I suggest to remove the Figure 2 from the main Manuscript, and to incorporate the current Figure 2 as Supplementary Material file (PDF, DOC, TIF, PNG, or another format). Then, the authors must renumber the Figures and check the text of the main Manuscript, as well as the new Figure S1 of the Supplementary Material file. Best regards.
Re: The chromatograms of Figure 2 was replaced by HPLC fingerprints of NIN and SIN, wihch looks clearer and more intuitive.